# Integrated Metabolomic and Transcriptomic Analysis of Puerarin Biosynthesis in *Pueraria montana* var. thomsonii at Different Growth Stages

**DOI:** 10.3390/genes14122230

**Published:** 2023-12-18

**Authors:** Xinyi Hu, Ting Zhu, Xinyi Min, Jianing He, Cong Hou, Xia Liu

**Affiliations:** School of Chemistry, Chemical Engineering and Life Sciences, Wuhan University of Technology, Wuhan 430070, China; 18963951698@163.com (X.H.); sixtnn@163.com (T.Z.); 17396177935@163.com (X.M.); 294016@whut.edu.cn (J.H.); hchappy1999@foxmail.com (C.H.)

**Keywords:** *Pueraria montana* var. thomsonii, puerarin biosynthesis, transcriptome, metabolome

## Abstract

Puerarin, a class of isoflavonoid compounds concentrated in the roots of *Puerarias*, has antipyretic, sedative, and coronary blood-flow-increasing properties. Although the biosynthetic pathways of puerarin have been investigated by previous researchers, studies focusing on the influence of different growth stages on the accumulation of metabolites in the puerarin pathway are not detailed, and it is still controversial at the last step of the 8-*C*-glycosylation reaction. In this study, we conducted a comprehensive analysis of the metabolomic and transcriptomic changes in *Pueraria montana* var. thomsonii during two growing years, focusing on the vigorous growth and dormant stages, to elucidate the underlying mechanisms governing the changes in metabolite and gene expression within the puerarin biosynthesis pathway. In a comparison of the two growth stages in the two groups, puerarin and daidzin, the main downstream metabolites in the puerarin biosynthesis pathway, were found to accumulate mainly during the vigorous growth stage. We also identified 67 common differentially expressed genes in this pathway based on gene expression differences at different growth stages. Furthermore, we identified four candidate 8-*C*-GT genes that potentially contribute to the conversion of daidzein into puerarin and eight candidate 7-*O*-GT genes that may be involved in the conversion of daidzein into daidzin. A co-expression network analysis of important UGTs and HIDs along with daidzein and puerarin was conducted. Overall, our study contributes to the knowledge of puerarin biosynthesis and offers information about the stage at which the 8-*C*-glycosylation reaction occurs in biosynthesis. These findings provide valuable insights into the cultivation and quality enhancement of *Pueraria montana* var. thomsonii.

## 1. Introduction

*Pueraria montana* var. thomsonii (Bentham) M. R. Almeida (hereafter referred to as *P. montana* var. thomsonii) is a perennial climbing vine of the Leguminosae family [1,2]. The Pueraria genus comprises more than 20 species, mainly distributed in temperate and subtropical regions [2,3]. *P. montana* var. thomsonii is a notable medicinal plant known for its pharmacological activities, including heart and brain protection, hepatoprotection, and hypotensive and hypoglycemic properties, as well as antitumor capabilities [4]. It contains various chemical compounds, such as isoflavonoids, triterpenoids, alkaloids, saponins, coumarins, and polysaccharides [5]. The primary medicinal compounds in the root tubers of *P. montana* var. thomsonii are flavonoids and isoflavones, including daidzein, genistein, daidzin (7-*O*-glycoside of daidzein), puerarin (8-*C*-glycoside of daidzein), etc. [6]. In addition to being used as a medicine, *P. montana* var. thomsonii is also in great demand as a food and nutraceutical, and it is popular in Asian countries, especially Japan and Thailand [7].

In recent years, omics technology has developed rapidly, and the integrated analysis of multi-omics data has been fruitfully used to study secondary metabolic components, identify gene functions, and elucidate metabolic pathways during plant development [8,9,10,11,12]. Studies on the biosynthesis of puerarin, an important secondary metabolite in *P. montana* var. thomsonii, have also been in full swing. Previous molecular biological studies on leguminous species have served as important references for exploring the potential biosynthetic pathways of puerarin in *P. montana* var. thomsonii [13,14]. Puerarin biosynthesis occurs through the phenylpropanoid and isoflavonoid hybrid pathway. Different flavonoids in plants share a common upstream phenylpropanoid pathway, which has been well studied [15,16]. In the phenylpropanoid pathway, phenylalanine ammonialyase (PAL), cinnamate 4-hydroxylase (C4H), 4-coumarate-CoA ligase (4CL), chalcone synthase (CHS), chalcone reductase (CHR), and chalcone isomerase (CHI) act sequentially as catalysts [16,17,18], while the isoflavonoid pathway is catalyzed sequentially via 2-hydroxyisoflavanone (IFS), also named 2-hydroxyisoflavanone synthase (2-HIS); 2-hydroxyisoflavanone dehydratase (HID); and UDP-glucosyltransferases (UGTs). The production of puerarin can occur through the direct action of 8-C-glucosyltransferase (8-C-GT) on daidzein or via an alternative route where the C-glycosyl linkage is introduced to the chalcone isoliquiritigenin [1,9,19,20,21,22]. Among them, the widely studied route is the daidzein pathway [22]. However, the stage at which the 8-C-glycosylation reaction occurs in the biosynthesis of puerarin remains controversial [23,24]. 

Plant development is a complex process, and the metabolites accumulated in plants during different developmental stages vary greatly [25]. However, most of the previous studies have focused on different tissues or different varieties to identify candidate genes from differentially expressed genes, and few studies have been conducted to investigate the transcriptomic and metabolomic profiles of *P. montana* var. thomsonii at different growth stages to probe the accumulation mechanisms of isoflavonoids in the puerarin biosynthesis pathways from the perspective of growth stages.

In this study, the differences in metabolites and genes in *P. montana* var. thomsonii root tubers during vigorous growth and dormant stages were investigated using ultraperformance liquid chromatography–tandem mass spectrometry (UPLC-MS/MS) and RNA sequencing (RNA-Seq) analysis techniques. Root tubers from one-year-old and two-year-old *P. montana* var. thomsonii from two growth stages were compared separately to obtain differentially expressed genes and differentially accumulated metabolites associated with growth stages that were common to both the one- and two-year-old sample sets, thus excluding other interferences and obtaining more reliable data. A detailed analysis of the gene expression data and phylogenetic relationship allowed us to propose important UGTs, including several C-UGT candidates and O-UGT candidates, that are likely involved in the downstream biosynthesis and regulation of puerarin and daidzin in *P. montana* var. thomsonii. It also offers information about the stage at which the 8-C-glycosylation reaction occurs during biosynthesis. In conclusion, we systematically investigated the metabolite accumulation and molecular mechanisms of puerarin biosynthesis during different growth stages of the root tuber of *P. montana* var. thomsonii using metabolomics and transcriptomics approaches combined with co-expression analysis, which provided new insights into the regulation of isoflavonoids and established a foundation for quality enhancement of *P. montana* var. thomsonii.

## 2. Materials and Methods

### 2.1. Plant Materials

The samples used in this study were collected from healthy *P.montana* var. thomsonii plants growing in Zhongxiang City, Hubei Province, China (112°54′33″ E, 31°21′12″ N, with an altitude of 154.3 m). Fresh root tubers from two growth years at two growth stages were separately gathered. The basic information of the samples and the content of puerarin are shown in Table 1. “GX” is defined as the vigorous growth stage (September) group, “GD” as the dormant stage (January) group, and the numbers following represent the growth years. For each growth stage at which the plant materials were sampled, three biological replicates with three technical repeats were performed, totaling 12 sets of samples. A total of 3 g of each set of samples was collected in EP tubes, frozen on dry ice, stored at −80°C until RNA, and metabolites were extracted. The remaining samples were dried and broken into a powder that could pass through a No. 3 sieve, which was subsequently made into a test solution to determine the content of puerarin using high-performance liquid chromatography.

### 2.2. Processing of Samples and Procedures of HPLC

The chromatograph was a Shimadzu L-16, and the column was manufactured by Dalian Elite Analytical Instruments Co., Ltd., Dalian, China. Octadecylsilane-bonded silica gel was used as filler; methanol–water (25:75) was used as the mobile phase; the detection wavelength was 250 nm.

In detail, a sample powder amount of about 0.8 g was taken, weighed precisely, put in a stoppered conical flask, mixed with 50 mL of 30% ethanol, and heated under reflux for 30 min; after cooling, it was filtered, and the filtrate was used to make the test solution.

Puerarin standard was purchased from Lemeitan Pharmaceuticals, and 30% ethanol was added to make a solution containing 80 μg per 1 mL, which served as the control solution.

A volume of 10 μL of both the control solution and test solution was injected into the liquid chromatograph, and determination was carried out [26].

### 2.3. RNA Extraction and RNA Sequencing

Ethanol precipitation and CTAB-PBIOZOL were used for extraction, and the successfully extracted RNA was dissolved by adding 50 µL of DEPC-treated water. Subsequently, total RNA was identified and quantified using NanoDrop and Agilent 2100 Bioanalyzer (Thermo Fisher Scientific, Waltham, MA, USA). After constructing the mRNA library, sequencing was performed with the Illumina platform. The downstream data were filtered to obtain clean data, and sequence comparison was performed with the reference genome to obtain mapped data, which were subsequently used to perform structural-level analysis, such as variable splicing analysis, new gene discovery gene structure optimization, and expression-level analysis, such as differential expression analysis, functional annotation of differentially expressed genes, and functional enrichment, based on the expression of the genes in different samples or different groups of samples. Our study was based on the genomic data of *P. montana* var. thomsonii from the Guangxi Academy of Agricultural Sciences, China, as a reference (GenBank ID: GCA_019096045.1).

### 2.4. UPLC–MS/MS-Based Widely Targeted Metabolomic Analysis

The samples were placed in a lyophilizer (Scientz-100F, Scientz, Ningbo, China) for vacuum freeze drying and then ground for 1.5 min to powder using a grinder (MM 400, Retsch, Haan, German) at 30 Hz. Then, 50 mg of powder was weighed using an electronic balance (MS105DM), and 1200 μL of pre-cooled 70% methanol aqueous internal standardized extract at −20 °C was added. The extract was vortexed once every 30 min for 30 s for a total of 6 times and centrifuged at 12,000 rpm for 3 min. The supernatant was aspirated, and the sample was filtered through a microporous filter membrane (0.22 μm pore size) and preserved in the injection vial for UPLC-MS/MS analysis.

Liquid phase detection was performed on an Agilent SB-C18 1.8 µm, 2.1 mm × 100 mm column, with solvent A (pure water with 0.1% formic acid) and solvent B (acetonitrile with 0.1% formic acid). The flow rate was 0.35 mL/min, the column temperature was 40 °C, the injection volume was 2 μL, and the elution was based on a gradient. The mass spectrometry conditions were as follows: electrospray ionization (ESI) temperature, 500 °C; ion spray voltage (IS), 5500 V (positive ion mode)/−4500 V (negative ion mode). The ion source gas I (GSI), gas II (GSII), and curtain gas (CUR) were set to 50, 60, and 25 psi, respectively, and the collision-induced ionization parameter was set to high. QQQ scans were performed using multiple reaction monitoring (MRM) mode with the collision gas (nitrogen) set to medium. DP and CE of individual MRM ion pairs were accomplished by further de-clustering potential (DP) and collision energy (CE) optimization. A specific set of MRM ion pairs was monitored in each period based on the metabolites eluted within each period.

The quantification of metabolites was performed using triple quadrupole mass spectrometry. In MRM mode, the quadrupole first screens the precursor ions of the target substance and excludes ions corresponding to other molecular weight substances to preliminarily eliminate interference. Precursor ions are induced to ionize by colliding cells and then disintegrate to form many fragmented ions. Then, the fragment ions are filtered through a triple quadrupole to select the desired characteristic fragments, eliminate interference from non-target ions, and make quantification more accurate and repeatable. After obtaining spectral analysis data of metabolites from different samples, peak area integration was performed on all mass spectra peaks of the substance, and integration correction was performed on mass spectra peaks of the same metabolite in different samples [27]. To compare the differences in the content of each metabolite in different samples among all detected metabolites, we corrected the mass spectrometry peaks detected for each metabolite in different samples based on the information on metabolite retention time and peak shape to ensure the accuracy of quantification and quantification. The relative content of metabolites in each sample was expressed as the integral of the chromatographic peak area.

### 2.5. Analysis of Differential Accumulation Metabolites and Differentially Expressed Genes

The variable importance in projection (VIP) value indicated the degree of influence of the intergroup difference of the corresponding metabolite in the classification discrimination of the samples in each group in the model, and the metabolites with VIP ≥ 1 were generally considered to have significant differences. Additionally, metabolites with a fold change (FC) greater than or equal to 2 or less than or equal to 0.5 between the control and experimental groups were considered significantly different. Overall, metabolites with VIP ≥ 1 and |log2 FC| ≥ 1 were defined as differentially accumulated metabolites (DAMs). 

Moreover, differential expression analysis between sample groups was performed using DESeq2 to obtain the set of differentially expressed genes between the two comparison groups, and the false discovery rate (FDR) was obtained via multiple hypothesis testing corrected for the probability of hypothesis testing (*p*-value) using the Benjamini–Hochberg method. Differentially expressed genes (DEGs) were screened for |log2FC| ≥ 1 and FDR < 0.05. 

### 2.6. Kyoto Encyclopedia of Genes and Genomes Annotation and Enrichment Analysis

Identified metabolites were annotated using the Kyoto Encyclopedia of Genes and Genomes (KEGG) compound database (http://www.kegg.jp/kegg/compound/, accessed on 1 June 2023), and annotated metabolites were then mapped to the KEGG Pathway database (http://www.kegg.jp/kegg/pathway.html, accessed on 1 June 2023). Pathways with significantly regulated metabolites were then subjected to metabolite set enrichment analysis (MSEA), and their significance was determined via hypergeometric test *p*-values.

### 2.7. Identification of Structural Genes in the Puerarin Biosynthesis Pathway

To identify all the structural genes involved in this pathway, known protein sequences were retrieved from the NCBI protein database (https://www.ncbi.nlm.nih.gov/protein/, accessed on 15 June 2023). These sequences were then searched on Interpro (https://www.ebi.ac.uk/interpro/search/sequence/, accessed on 15 June 2023) to obtain the Pfam numbers and target structural domains. Then, a biosequence analysis using profile hidden Markov models (HMMERs) was used to search sequence databases for homologous sequences and perform sequence alignments. Simultaneously, we employed local BLAST 2.14.0 software to identify the target structural genes within the *P. montana* var. thomsonii transcriptome of our current study. The results obtained from HMMER and BLAST were combined, and duplicate sequences were removed. Sequences with incomplete conserved structural domains were further screened using the CD-search (https://www.ncbi.nlm.nih.gov/Structure/cdd/wrpsb.cgi, accessed on 20 June 2023) online protein structure prediction website. Finally, manual screening was conducted to ensure the selection of relevant sequences. 

### 2.8. Phylogenetic Analysis of UGTs in the Puerarin Biosynthesis Pathway

The glycosyltransferase (GT) protein superfamily contains numerous members, with family 1 (also known as ureido-diphosphate-glycosyltransferase UDP-glycosyltransferase (UGT)) being the largest and most closely related to plants [28]. Due to the diversity and specificity of glycosyl acceptors and glycosyl donors, UGT can be divided into O-glycosyltransferases (OGTs) and C-glycosyltransferases (CGTs), which can be further subdivided according to the modification sites of glycosylation reactions.

To perform a phylogenetic analysis of the UGTs, protein sequences of previously characterized UGTs with known structures and functions from *Pueraria montana* var. lobata (referred to as *P. montana* var. lobata) or other species were downloaded from the NCBI protein database (https://www.ncbi.nlm.nih.gov/protein/, accessed on 25 June 2023). All UGT sequences were aligned using the MUSCLE algorithm, and a phylogenetic tree was constructed using the neighbor-joining method with MEGA 11.0 software, with a bootstrap value of 1000. 

### 2.9. Quantitative Real-Time PCR

To validate the structural gene expression profiles, ABScript Neo RT Master Mix for quantitative real-time polymerase chain reaction (qRT-PCR) with gDNA Remover kit was used to reverse transcribe the sample RNA into cDNA. Subsequently, a BrightCycle Universal SYBR Green qPCR Mix with UDG was used for qRT-PCR analysis of DEGs using the Applied Biosystems^TM^ QuantStudio^TM^ 3&5 (Waltham, MA, USA) real-time quantitative PCR instrument. The following conditions were applied: 37 °C for 2 min, 95 °C for 3 min, and then 40 cycles of 95 °C for 5 s and 60 °C for 30 s, followed by a melt cycle of 95 °C for 5 s and 60 °C for 11 min. β-actin was selected as the internal reference gene [20]. 

Primers for qRT-PCR were designed using Primer Premier 5, ensuring amplicon lengths of 80–150 bp, GC contents of 40–60%, and Tm values of 50–60 °C. The primer sequences can be found in Appendix A. All qRT-PCR experiments were taken in three biological replicates, and each reaction was performed in triplicate. The relative expression of each gene was calculated with the 2^−ΔΔCt^ method [29].

## 3. Results

### 3.1. Determination of Puerarin in P. montana var. thomsonii at Different Growth Stages via HPLC

The contents of puerarin in 12 groups of samples were separately determined using HPLC. It was found that the content of puerarin was higher during the vigorous growth stage than during the dormant stage, both in one-year and two-year-old root tubers (Table 2). The raw data of the content determination and the related chromatograms are displayed in Appendix A.

### 3.2. Metabolomic Profiling and Differentially Accumulated Metabolite Enrichment of P. montana var. thomsonii at Different Growth Stages

To better understand the nutritional and medicinal differences among *P. montana* var. thomsonii in the two growth stages, widely targeted UPLC-MS/MS-based metabolite profiling of the samples was performed. A total of 1525 metabolites were detected based on the UPLC-MS/MS detection platform and a self-built database (Figure 1a), including 285 flavonoids, 126 lipids, 204 amino acids and derivatives, 152 alkaloids, 76 organic acids, 177 terpenoids, 55 nucleotides and derivatives, 57 lignans and coumarins, 143 lignans and coumarins, 20 quinones, 13 steroids, 4 tannins, and 105 others metabolites [30]. The detailed information of these metabolites is shown in Appendix A. Of the 285 flavonoids in the four groups, 10 chalcones, 24 flavanones, 3 flavanonols, 1 dihydroisoflavone, 6 anthocyanidins, 77 flavones, 39 flavanols, 96 isoflavones, and 29 other flavonoids were detected. The results of PCA analysis revealed a clear separation between the four groups of samples and quality control (QC) samples, and the repeated samples in each group were gathered, except sample GX2-3, which showed slight variation, indicating the repeatability and reliability of the experiments (Figure 1b). 

We used the supervised method, OPLS-DA, and Student’s *t*-test (value of *p* < 0.05) to find the metabolites responsible for differences among these four groups. In this study, the OPLS-DA model compared metabolite contents of the stages in pairs to evaluate the differences. The Q2 and R2 values of the two comparison groups exceeded 0.5, demonstrating that the models were stable [31] (Appendix A). The DAMs (|Log (FC)| > 1, VIP ≥ 1, and *p*-value < 0.05) between varieties were screened.

In terms of group comparisons, 281 DAMs in the GX1 vs. GD1 group were observed, with 65 upregulated and 216 downregulated metabolites. Similarly, the GX2 vs. GD2 group exhibited 184 DAMs, including 48 upregulated and 136 downregulated metabolites. In addition, based on KEGG enrichment analysis, the DAMs in the two comparison groups were both mainly enriched in “biosynthesis of secondary metabolites” and “isoflavonoid biosynthesis” (Figure 1c).

### 3.3. Transcriptomic Profiling and Differentially Expressed Gene Screening of P. montana var. thomsonii at Different Growth Stages

After testing the RNA quality, a total of 82.98G high-quality bases were generated, with GC bases accounting for 43.86−44.69% and Q30 bases exceeding 92.28%. The RNA integrity was good, and the total amount complied with the requirements for standard library construction. Transcriptome sequencing was performed. PCA separated the two samples in each comparison group clearly (Figure 2a).

Differential genes were screened for |log2FC| ≥ 1 and FDR < 0.05. A total of 16300 DEGs were obtained from four groups of samples. There were 7543 DEGs in the GX1 vs. GD1 group, with 3785 upregulated and 3758 downregulated genes. Similarly, the GX2 vs. GD2 group exhibited 7171 DEGs, including 3102 upregulated and 4069 downregulated genes. Detailed information is shown in Appendix A. The volcano map visualizes the overall distribution of differential genes in the two groups of samples (Figure 2b), indicating significant differences in gene expression levels during root development in *P. montana* var. thomsonii. To further analyze the DEGs, KEGG enrichment analysis was conducted, and the top 20 enriched pathways are shown in the bubble diagram. Notably, twelve of these pathways were shared by both comparison groups and primarily enriched in “metabolic pathways” and “starch and sucrose metabolism” (Figure 2c). 

In the transcriptome generated in this study, a total of 4479 transcripts were annotated as transcription factors (TFs), mainly belonging to the AP2/ERF-ERF, WRKY, MYB, bHLH, C2H2, SNF2, and C3H families (Appendix A). MYB transcription factors have been shown to regulate secondary metabolism, stress responses, and development in various plants [32]. Specifically, in *P. montana* var. thomsonii, they play a role in the biosynthesis of puerarin, where the expression of nine key enzymes is regulated by specific MYBs in the phenylpropanoid and isoflavonoid pathways [33]. A total of 261 MYBs were detected in this study.

### 3.4. Analysis of Metabolites and Differentially Expressed Genes Involved in the Puerarin Biosynthesis Pathway

Based on previous research by Shang et al. [21], a possible pathway map for puerarin biosynthesis was constructed. The conversion of phenylalanine to cinnamic acid is catalyzed by PAL, followed by the conversion of cinnamic acid to 4-coumarin coenzyme A through the actions of C4H and 4CL. Subsequently, CHS and CHR polymerize 4-coumarin coenzyme A to isoliquiritigenin. CHI then catalyzes the formation of liquiritigenin from isoliquiritigenin, which is further catalyzed by IFS to produce 2,7,4′-trihydroxyisoflavonone. Thereafter, chalcone isoflavone can be catalyzed by HID and 8-C-GT to produce the target compound, puerarin, via two different pathways (Figure 3).

Among all the metabolites, six metabolites were labeled in the puerarin biosynthesis pathway, which were puerarin, daidzin, daidzein, isoliquiritigenin, liquiritigenin, and L-phenylalanine [34]. The expression levels of these metabolites were analyzed using a heat map, as shown in Figure 3. As expected, in a two-by-two comparison, the downstream metabolites (puerarin and daidzin) accumulated more in the vigorous growth group, whereas the upstream compounds (isoliquiritigenin, liquiritigenin, and L-phenylalanine) accumulated more in the dormant group. In particular, differences in upstream metabolites were more pronounced in the one-year samples, while differences in downstream metabolites were more pronounced in the two-year samples.

Using profile hidden Markov models (HMMERs) and local BLAST 2.14.0 software, several key structural genes involved in the puerarin biosynthesis pathway were identified, including 12 PALs, 109 4CLs, 53 CHSs, 92 CHRs, 11 CHIs, 19 IFSs, 510 UGTs, and 91 HIDs. All DEGs were combined with all structural genes on the pathway to form an intersection, and the genes in common are likely to be differential structural genes involved in the puerarin biosynthesis pathway. In the GX1 vs. GD1 group, a total of 114 DEGs can be labeled on the pathway, including 3 PALs, 6 IFSs, 8 4CLs, 4 CHSs, 16 CHRs, 1 CHI, 70 UGTs, and 6 HIDs. Similarly, in the GX2 vs. GD2 group, a total of 118 DEGs can be labeled on the pathway, including 7 IFSs, 10 4CLs, 3 CHSs, 12 CHRs, 2 CHIs, 76 UGTs, and 8 HIDs. Notably, there were 67 common DEGs between the two groups, including 7 4CLs, 2 CHSs, 10 CHRs, 3 IFSs, 43 UGTs, and 2 HIDs. A heatmap of gene expression was derived based on FPKM values, as shown in Appendix A, and it is also labeled next to each step in Figure 3. As can be seen from the data, the expression of UGTs has a certain regularity. That is, the expression was higher at the vigorous growth stage (GX) than at the dormant stage (GD), and the expression was especially high at the vigorous growth stage during the second year (GX2). IFS was also expressed significantly higher at the vigorous growth stage than at the dormant stage. Similarly, HID was also expressed at high levels at the vigorous growth stage, but the differences were not as pronounced.

### 3.5. Phylogenetic Analysis of Candidate UGTs

To further evaluate their possible regulatory roles, a phylogenetic analysis was conducted on the 43 UGTs, along with the previously characterized UGTs of known structures and functions from other species [35,36]. Orthologous genes with similar functions are usually clustered in the same clades (Figure 4). The information on UGTs used in phylogenetic analysis is listed in Appendix A. Based on the phylogenetic tree topology, the proteins were clustered into nine clades. Only PtUGT35 was not assigned to any clades. Except for 14 UGTs clustered together that did not cluster with previously reported UGTs, the remaining 28 UGTs all clustered with different UGTs, which can be differentiated into 17 OGTs and 11 CGTs. A total of 17 OGTs can be further subdivided into 8 isoflavone 7-O glycosylations, 5 flavone 5-O glycosylations, and 4 flavone 7-O glycosylations. Notably, the eight isoflavone 7-O glycosylations (PtUGT1, PtUGT2, PtUGT3, PtUGT4, PtUGT7, PtUGT8, PtUGT32, and PtUGT36) clustered with 7-O-GT reported in *P. montana* var. lobata might be involved in catalyzing the production of daidzein from daidzein [37]. Four of the eleven CGTs (PtUGT33, PtUGT34, PtUGT38, and PtUGT39) clustered with PlUGT43, which is the first 8-C-GT proven to catalyze the generation of puerarin from daidzein in puerarin biosynthesis [22], and thus, it can be hypothesized that these four 8-C-GTs in the present study also have the same function, pending further verification. The other seven clustered with the 8-C-GT of Trollius chinensis [38]. The protein sequences of candidate UGTs are listed in Appendix A.

### 3.6. Co-Expression Analysis of Important Genes and Metabolites

In order to examine the relationship between important metabolites and genes related to puerarin biosynthesis in *P. montana* var. thomsonii at different growth stages, a co-expression network analysis of chosen metabolites and genes was conducted. Correlations were analyzed, and network diagrams were drawn based on the Pearson correlation coefficient, as well as the *p*-value (Figure 5). The color of the line indicates the correlation, where red is a positive correlation and blue is a negative correlation; the thickness of the line indicates the size of the *p*-value; and the metabolites and genes are differentiated by the shapes of the nodes. It can be observed that as an important intermediate, daidzein is strongly associated with almost all candidate UGTs and shared differential HID genes. Moreover, puerarin is tightly correlated with the four 8-C-GTs and less correlated with the two HIDs, which also corroborates the pathway by which daidzein is catalyzed by 8-C-GT to produce puerarin. The associations between daidzein and puerarin for these candidate UGTs are roughly consistent.

### 3.7. Quantitative Real-Time Polymerase Chain Reaction Validation of the Expression Levels of the Genes Associated with Puerarin Biosynthesis

To confirm the RNA-Seq results, the transcript abundance of eight genes in each of the two comparison groups was analyzed via qRT-PCR [39]. These genes included the main structural genes of the flavonoid biosynthetic pathway. The results showed that the expressions of the genes related to puerarin biosynthesis determined via qRT-PCR were consistent with the corresponding FPKM values obtained from the RNA-Seq analysis (Figure 6). 

## 4. Discussion

In recent years, the advancement of metabolomics and transcriptomics has provided new approaches for biosynthetic mechanisms of secondary metabolites in medicinal plants. Therefore, further studies have been conducted on the puerarin biosynthesis pathway in Pueraria [1,9,14,19,40]. While the upstream of the pathway has been well characterized, the enzymes responsible for synthesizing the key downstream metabolite puerarin are still controversial and subject to some contradictions. Moreover, most of these studies focus on comparing transcriptomes of different tissue parts, such as roots, stems, and leaves, or comparing different varieties to identify candidate genes that play roles in the puerarin biosynthesis pathway. However, few studies of *P. montana* var. thomsonii have been conducted to investigate the comparative transcriptomic and metabolomic studies to probe the differential expression of structural genes and accumulation of metabolites on puerarin biosynthesis pathways from the perspective of growth stages.

The upstream phenylalanine pathway of the puerarin biosynthesis has been well studied, but the downstream isoflavonoid pathway remains controversial, especially the stage at which the 8-C glycosylation reaction occurs. Previous research has suggested two possible pathways. Puerarin can occur via a route where the C-glycosyl linkage is introduced to the chalcone isoliquiritigenin or through the direct action of 8-C-GT on daidzein. However, no differential expression data of the puerarin precursor 2,7,4′-trihydroxyisoflavanone 8-C-glycosyl, which supported the first pathway, were obtained in our study. However, we detected differential accumulation data of daidzein, which further validated the introduction of 8-C glycosyl into the daidzein pathway. Additionally, in our study, we found some common patterns in the accumulation of the metabolites on the puerarin pathways. Differences in the accumulation of upstream and downstream metabolites in the pathway at different growth stages then lead to the regularity of accumulation stages for isoflavone biosynthesis. The accumulation of upstream metabolites was mainly accomplished during the dormant stage, while the accumulation of downstream target products occurred mainly during the vigorous growth stage. Moreover, it is worth noting that the difference in the accumulation of upstream compound metabolites was more obvious in the comparison of the GX1 vs. GD1 group, and the difference in the accumulation of downstream target compounds was more obvious in the comparison of the GX2 vs. GD2 group, so it can be speculated that the upstream metabolites that accumulated in the first year were consumed in the second year to obtain the downstream target products. Therefore, this study also provides an information basis for further exploring and verifying the impact of age on the puerarin biosynthesis pathway.

Glycosyltransferase (GT) is a superfamily of proteins that catalyze glycosylation modification reactions, which can alter the solubility, stability, and other properties of the substrate to make the substrate molecule more versatile [41]. Among all the GT families, the UGT family is most closely related to the downstream metabolites puerarin and daidzin. In our study, the 43 PtUGTs from both comparison groups were used for phylogenetic analysis. The results revealed that eight 7-O-GTs might be involved in the catalytic synthesis of daidzin from daidzein; in addition, four 8-C-GTs (PtUGT33, PtUGT 34, PtUGT 38, and PtUGT 39) clustered closely with the previously reported PlUGT43, which demonstrates that they might play an important role in catalyzing the generation of puerarin from daidzein [22,37]. Therefore, these four UGTs could be the most likely candidates. It is further evidence of the authenticity of the pathway that the C- glycosyl linkage is introduced to daidzein to produce puerarin. However, their function cannot be determined based on sequence alone, and whether they play a role in puerarin biosynthesis remains to be further verified.

All these studies contribute to the knowledge of puerarin biosynthesis and offer information about the stage at which the 8-C-glycosylation reaction occurs in the pathway. These findings provide valuable insights into the cultivation and quality enhancement of *Pueraria montana* var. thomsonii.

## 5. Conclusions

In our study, *P. montana* var. thomsonii from two growing years, at both the vigorous and dormant growth stages, were selected to explore the underlying mechanism of the puerarin biosynthesis pathway, from both phenotypic and genetic perspectives. The results of the metabolic analysis showed that a total of 1525 metabolites were detected and quantified, of which flavonoids were dominant. Transcriptomic analysis showed that a total of 16,300 DEGs were identified, of which 67 DEGs were common between the two groups in the puerarin biosynthesis pathway. The accumulation stage of the major metabolites in the pathway was investigated, and we found that the accumulation of upstream metabolites is mainly accomplished during the dormant stage, whereas the accumulation of downstream target products occurs mainly during the vigorous growth stage. This research provides new guidance for the cultivation of *P. montana* var. thomsonii. The genes involved in the isoflavone biosynthesis pathway were further refined in our studies, identifying four 8-C-GTs that might be able to act on daidzein to generate puerarin and eight 7-O-GTs that might be able to act on daidzein to generate daidzin. These findings further validated the authenticity of the daidzein pathway to obtain puerarin. The integrated transcriptomic and metabolomic analyses provide evidence for the C-glycosylation stage of the puerarin biosynthetic pathway and the mechanism of metabolite accumulation and also provide a reference for the quality enhancement of *P. montana* var. thomsonii.

## Figures and Tables

**Figure 1 genes-14-02230-f001:**
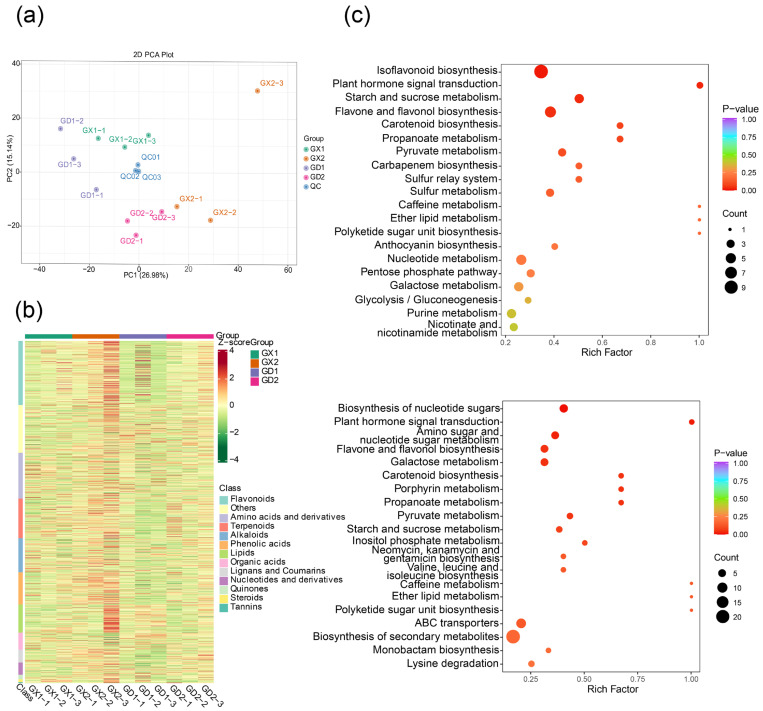
Metabolite accumulation of *P. montana* var. thomsonii at different growth stages. (**a**) Metabolite accumulation at different growth stages. (**b**) Principal component analysis (PCA) of metabolites in different groups. (**c**) KEGG analysis of all DAMs: the upper section represents the results of the GX1 vs. GD1 group, while the lower section represents the results of the GX2 vs. GD2 group. Each bubble in the plot represents a metabolic pathway, the abscissa and bubble size of which jointly indicate the magnitude of the impact factors of the pathway. A larger bubble size indicates a larger impact factor. The abscissa of bubbles indicates the enrichment ratio of metabolites in each pathway. The bubble colors represent the *p*-value of the enrichment analysis, with lighter colors showing a higher confidence level. Sorted by the *p*-value, the top 20 metabolic pathways are plotted in the bubble chart.

**Figure 2 genes-14-02230-f002:**
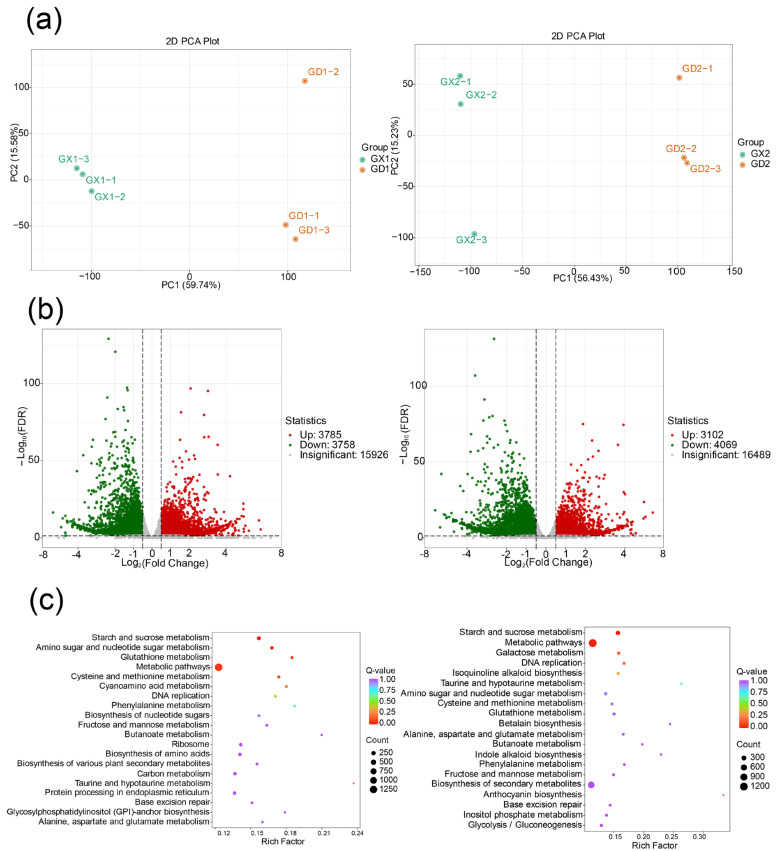
Analysis of differentially expressed genes (DEGs) of *P.montana* var. thomsonii at different growth stages: the left side of the figure represents the GX1 vs. GD1 group, while the right side represents the GX2 vs. GD2 group. (**a**) Principal component analysis (PCA) of the two comparison groups. (**b**) Differential gene volcano maps for the two comparison groups. (**c**) KEGG analysis of all DEGs for the two comparison groups. Each bubble in the plot represents a pathway, the abscissa and bubble size of which jointly indicate the magnitude of the impact factors of the pathway. A larger bubble size indicates a larger impact factor. The abscissa of bubbles indicates the enrichment ratio of genes in each pathway. The bubble colors represent the *Q*-value of the enrichment analysis, with lighter colors showing a higher confidence level. Sorted by the *Q*-value, the top 20 metabolic pathways are plotted in the bubble chart.

**Figure 3 genes-14-02230-f003:**
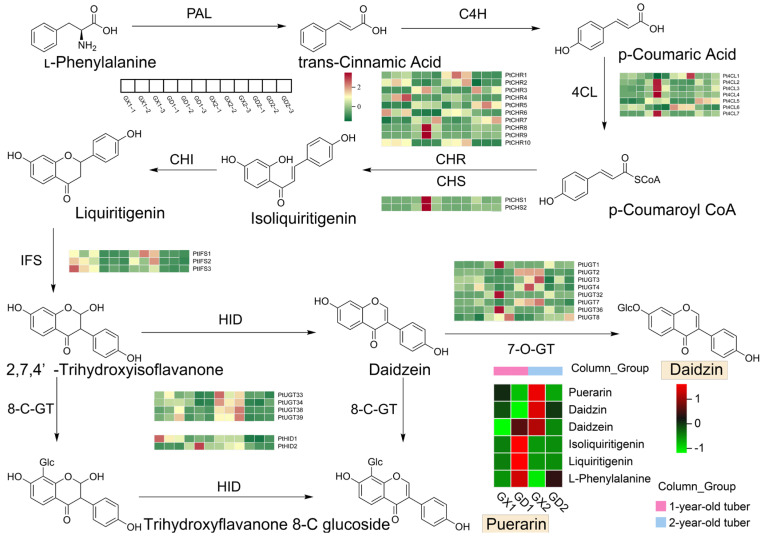
Puerarin biosynthetic pathways in *P. montana* var. thomsonii. DEG heatmaps are labeled at each step, and the DAM heatmap is labeled in the lower right corner.

**Figure 4 genes-14-02230-f004:**
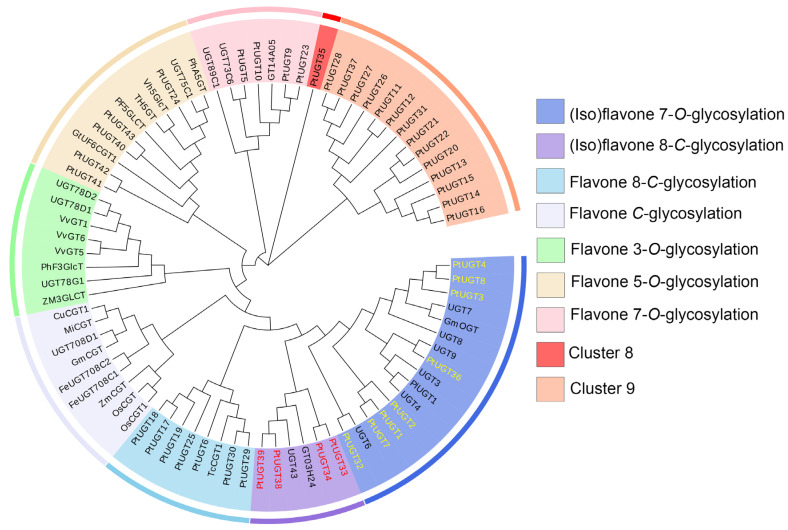
Phylogenetic analysis of 43 differential UGT genes shared by the two comparison groups. The red-labeled UGTs are highly homologous to 8-C-GT UGT43 in *P. montana* var. lobata, and yellow-labeled UGTs are highly homologous to 7-O-GT PlUGT1 in *P. montana* var. lobata.

**Figure 5 genes-14-02230-f005:**
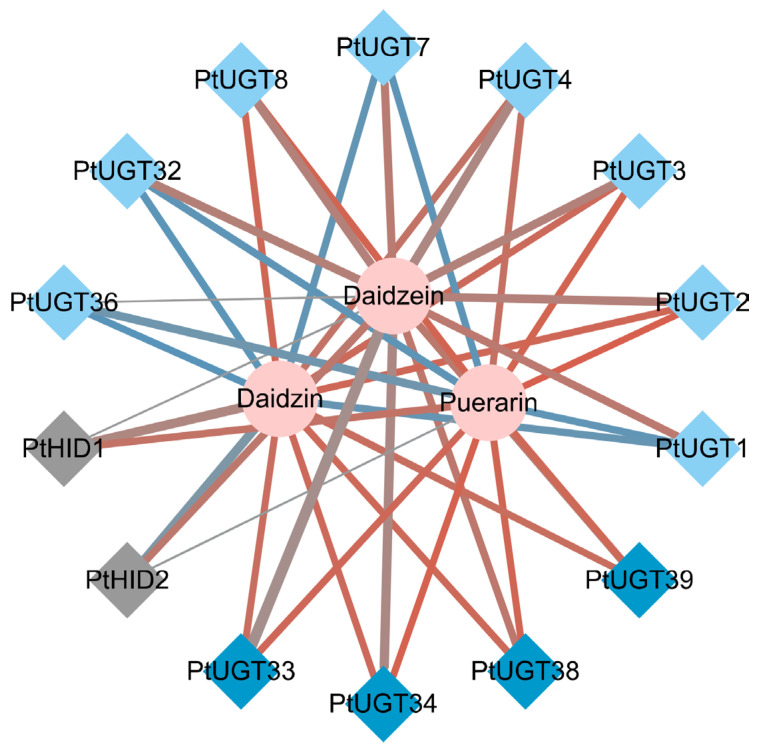
Interaction network of important differential structural genes and metabolites involved in puerarin biosynthesis. Circles indicate metabolites, diamonds indicate structural genes, and different genes are distinguished by different colors, where 8-C-GT is dark blue, 7-O-GT is light blue, and HID is gray.

**Figure 6 genes-14-02230-f006:**
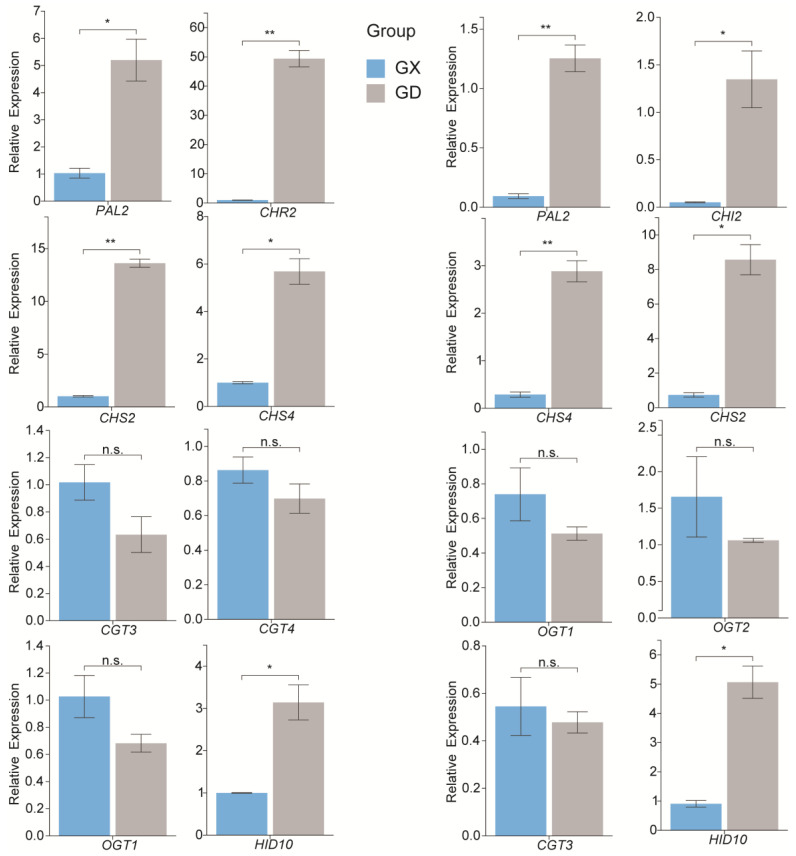
qRT-PCR validation of the genes associated with puerarin biosynthesis in *P. montana* var. thomsonii at different growth stages. The left is the one-year-old tuber comparison group, and the right is the two-year-old tuber comparison group. Values are means ± SD of three biological replicates (*n* = 3). ** *p* < 0.01, * *p* < 0.05, n.s.—not significant.

**Table 1 genes-14-02230-t001:** Sample information of *P. montana* var. thomsonii.

Sample	Describe	Excavation Time
GX1	One-year-old root tubers	September 2022
GX2	Two-year-old root tubers	September 2022
GD1	One-year-old root tubers	January 2023
GD2	Two-year-old root tubers	January 2023

**Table 2 genes-14-02230-t002:** Puerarin content connected with sample information.

Sample	Describe	Excavation Time	Puerarin Content (g/Mean ± SD)
GX1	One-year-old root tubers	September 2022	0.44 ± 0.05
GX2	Two-year-old root tubers	September 2022	0.96 ± 0.11
GD1	One-year-old root tubers	January 2023	0.06 ± 0.00
GD2	Two-year-old root tubers	January 2023	0.38 ± 0.03

## Data Availability

The original contributions presented in this study are included in this article/in the Appendix A, and further inquiries can be directed to the corresponding author.

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
