# Peer review of "Integrated Metabolomic and Transcriptomic Analysis of Puerarin Biosynthesis in Pueraria montana var. thomsonii at Different Growth Stages"

_genes, 2023, doi:10.3390/genes14122230_

Round 1
Reviewer 1 Report
Comments and Suggestions for Authors
The submitted manuscript “Integrated metabolomic and transcriptomic analysis of puerarin biosynthesis in Pueraria montana var. Thomsonii in different growth stages” [genes-2746081] written by Xinyi Hu, Ting Zhu, Xinyi Min, Jianing He, Cong Hou, Xia Liu describes a detailed transcriptomic and metabolomic study of Pueraria montana var. thomsonii with respect to the metabolite and gene expression within the puerarin biosynthesis. The practical, analytical and theoretical work is very well planned and performed. All investigations are performed with modern and/or common state of the art methods. The results and the conclusions are perspicuous. In making this assessment, the reviewer refers in particular to metabolomic analysis, since this is her/his main area of expertise. In the transcriptomic examination, however, she/he did also not notice any problems. The overall work is well planned and performed.
The results possess importance in furthering our knowledge about the entire biochemical pathway in Pueraria montana on transcriptomic and metabolomic level. The manuscript is hence of interest in the fields of Genetic, Phytochemistry, and to some extent in pharmaceutical sciences. It is close to a form to be published as research Article in “Genes”. Therefore, there are only a few minor comments, which should be considered by the authors and be taken into account prior to the acceptance of the manuscript.
Minor comments:
a) The reviewer assumes, that the plant was exactly “Pueraria montana var. thomsonii (Benth.) Wiersema ex D.B. Ward”. If this is correct, it should be mentioned in the manuscript one time.
b) The font size within the figures is very small and often difficult to read. Authors are encouraged to make this a little more readable.
c) The stereochemistry indicators of Fischer nomenclature (“D” and “L”) should be used in small caps rather than upper case throughout the manuscript.
d) There are very few typing mistakes, which should be corrected to avoid misunderstanding. E.g.:
line 3: “montana” not with capital:
line 39: “7-O-glucoside” with “O” in italics and without blank;
line 79: (and elsewhere): P.m.thomsonii with both “P” and “m” in italics;
lines 219-220: there are some blancs behind commas missing
e) Actually a task for the typesetter: The journal titles in the references should be spelled out or abbreviated throughout in accordance with the guidelines of Genes.
Author Response
Thanks for your opinions. These comments are very helpful to improve the quality of the manuscript. We have carefully revised our manuscript. Now I response the your comments in a point-to-point way. We sincerely hope that you find our responses and modifications satisfactory and that the manuscript is now acceptable for publication.
The ttachments contain revised tables and figures as well as the manuscript. We have two versions of the manuscript, one of which is the revised and completed version, and the other has the revised parts marked in yellow.
Best wishes,
Xia Liu

Reviewer 2 Report
Comments and Suggestions for Authors
Manuscript number: genes-2746081
Manuscript title: Integrated Metabolomic and Transcriptomic Analysis of Puerarin Biosynthesis in Pueraria Montana var. Thomsonii in Different Growth Stages
Authors: Xinyi Hu, Ting Zhu, Xinyi Min, Jianing He, Cong Hou, and Xia Liu
The topic treated in the present manuscript is of potential interest for the Journal readership, the amount of the experimental work done is considerable, the methodology appears to be adequate, even if more details should be given concerning the analysis of puerarin (see below). The statistical treatment of the experimental data seems to be adequate as well. The manuscript is also reasonably well written, linguistically speaking.
This being said, however, I found the present manuscript very difficult to follow, and hence to evaluate, especially as far the M&M and Results sections are concerned. So that, I would encourage the Authors to re-write it, in a much more clear, plain, concise, focused, essential manner, paying a special attention to the following aspects:
- the rationale and the novelty of the present study, on the basis of previous work on the same subject (the plant species and the metabolite of interest)
- the experimental design
- the procedure used for the quantitative analysis of puerarin, including extraction and usage of authentic standards
- the experimental proofs obtained here, beyond correlative assumptions and literature data, strictly concerning the very core of the present study, i.e. the biosynthesis of puerarin in Pueraria montana
In doing the above, the Authors should stay strictly focused on the very object of the present work, i.e. exclusively on the results which are pertinent to it, and should avoid the compulsive, pervasive and confusing use of acronyms and jargon, to allow the reader to fully appreciate the step forward made by the present experiments.
All the above considering, I recommend major revision of the present manuscript, properly and carefully addressing all the points raised above.
Comments on the Quality of English Language
Moderate editing of English language required
Author Response

(The authors gave the same response as above.)

Round 2
Reviewer 2 Report
Comments and Suggestions for Authors
Manuscript number: genes-2746081-v2
Manuscript title: Integrated Metabolomic and Transcriptomic Analysis of Puerarin Biosynthesis in Pueraria montana var. Thomsonii in Different Growth Stages
Authors: Xinyi Hu, Ting Zhu, Xinyi Min, Jianing He, Cong Hou, and Xia Liu
In the present revised version, I appreciated the sincere and substantial effort made by the Authors to accommodate suggestions and criticisms raised from my side on their original submission.
So, please find below some minor editorial suggestions:
1) I would not use throughout the expression “in different growth stages”….Rather “at different growth stages” or “during different growth stages”
2) HPLC is one of those quite common acronyms, within the scientific field, which does not need to be spelled out
3) It is usual to abbreviate the genus name, after it first mention in full, e.g. P. montana, whereas it is quite unusual to abbreviate the name of the species. So please do not use P.m. as the abbreviation for P. montana
4) Please never use acronyms in the titles of the subsections (e.g. “3.2 Metabolomic Profiling and DAMs Enrichment of P.m.thomsonii in Different Growth Stages” or “3.5 Phylogenetic Analysis of Candidate UGTs”
5) Please correct the spelling of the acronym DAMs: “differentially accumulated metabolites” is much better and less ambiguous than “differential accumulation metabolites (DAMs)”
6) Please use capital “L” as the abbreviation of liter, i.e. “mL” and not “ml”
7) Please check the presence of spaces among words, which often are omitted…e.g “50 mL” and not “50ml”
All the above considering, I recommend minor revision of the present manuscript, not requiring further examination from my side.
Comments on the Quality of English Language
Minor editing of English language required
Author Response
Thank you for your new suggestions and affirmation of the previous modified version. We have also provided point-to-point responses to the new questions you raised. We sincerely hope that you find our responses and modifications satisfactory and that the manuscript is now acceptable for publication.
The attachments contain revised tables and figures as well as the manuscript. We have two versions of the manuscript, one of which is the revised and completed version, and the other has the revised parts marked in yellow.
Best wishes,
Xia Liu
